# Effect of Natural Deep Eutectic Solvents on *trans*-Resveratrol Photo-Chemical Induced Isomerization and 2,4,6-Trihydroxyphenanthrene Electro-Cyclic Formation

**DOI:** 10.3390/molecules27072348

**Published:** 2022-04-06

**Authors:** Roberto Mattioli, Daniel Di Risola, Rodolfo Federico, Alessia Ciogli, Francesco Gasparrini, Claudio Villani, Mario Fontana, Anna Maggiore, Maria d’Erme, Luciana Mosca, Antonio Francioso

**Affiliations:** 1Department of Biochemical Sciences “A. Rossi Fanelli”, Sapienza University of Roma, p.le Aldo Moro 5, 00185 Rome, Italy; roberto.mattioli@uniroma1.it (R.M.); daniel.dirisola@uniroma1.it (D.D.R.); mario.fontana@uniroma1.it (M.F.); anna.maggiore@uniroma1.it (A.M.); maria.derme@uniroma1.it (M.d.); 2Active-Italia S.r.l., Via delle Terme Deciane 10, 00153 Rome, Italy; federico@active-italia.com; 3Department of Chemistry and Technology of Drugs, Sapienza University, p.le Aldo Moro 5, 00185 Rome, Italy; alessia.ciogli@uniroma1.it (A.C.); francesco.gasparrini@uniroma1.it (F.G.); claudio.villani@uniroma1.it (C.V.)

**Keywords:** *trans*/*cis*-resveratrol, photochemistry, 2,4,6-trihydroxyphenanthrene, isomerization and 6π photochemical electrocyclization, chromatographic separation, pentafluorophenyl stationary phase, natural deep eutectic solvents, photo-stability, UV light

## Abstract

*trans*-Resveratrol is a natural bioactive compound with well-recognized health promoting effects. When exposed to UV light, this compound can undergo a photochemically induced *trans*/*cis* isomerization and a 6π electrochemical cyclization with the subsequent formation of 2,4,6-trihydroxyphenanthrene (THP). THP is a potentially harmful compound which can exert genotoxic effects. In this work we improved the chromatographic separation and determination of the two resveratrol isomers and of THP by using a non-commercial pentafluorophenyl stationary phase. We assessed the effect of natural deep eutectic solvents (NaDES) as possible photo-protective agents by evaluating *cis*-resveratrol isomer and THP formation under different UV-light exposure conditions with the aim of enhancing resveratrol photostability and inhibiting THP production. Our results demonstrate a marked photoprotective effect exerted by glycerol-containing NaDES, and in particular by proline/glycerol NaDES, which exerts a strong inhibitory effect on the photochemical isomerization of resveratrol and significantly limits the formation of the toxic derivative THP. Considering the presence of resveratrol in various commercial products, these results are of note in view of the potential genotoxic risk associated with its photochemical degradation products and in view of the need for the development of green, eco-sustainable and biocompatible resveratrol photo-stable formulations.

## 1. Introduction

Resveratrol (3,5,4-trihydroxy-*trans*-stilbene) is a natural compound present in different agricultural and food products (e.g., peanuts, grapes, red wine) endowed with multiple pharmacological properties such as antioxidant, anti-inflammatory, cardiovascular, neuroprotective, chemo-preventive, antiviral and anti-aging activities [1,2,3,4]. However, in spite of its well-documented beneficial effect on human health, resveratrol use as a drug is strongly limited by its poor solubility and low bioavailability, and for its tendency to be unstable due to auto-oxidation and photochemical degradation, features that limit its use in topic and liquid formulations [5]. This compound is present in nature in its more stable isomeric form, the *trans*- form, that is favorite with respect to the *cis*- one. The *trans*- form can be converted into the *cis*- form by exposure to UV-light irradiation or direct sunlight [6,7]. However, after irradiation of *trans*-resveratrol, further reactions may occur other than the *trans*/*cis* isomerization, and different chemical species can be formed that can have a different biological activity than *trans*-resveratrol. As a matter of fact, the *trans*-resveratrol photodegradation pathway includes quinonoid and reactive radical species, whose identification is often elusive to spectroscopic and spectrometric investigations [7,8]. After UV-induced isomerization, the *cis*- form can undergo a 6π photochemical electrocyclization with subsequent oxidation to form a phenanthrenoid product previously characterized in our laboratory as 2,4,6-trihydroxyphenanthrene (THP, Figure 1). The phenanthrenic ring formation is a formal 6π electrochemical cyclization and takes place when the triene system of *cis*-resveratrol undergoes a (4n + 2)-electron conrotatory pericyclic ring closure with subsequent oxidation and generation of the fully aromatic tricyclic system of THP [9].

The polycyclic aromatic nature of this photo-derivative suggested a possible harmful effect for human health. In a recent study, Francioso et al. (2019) demonstrated that THP can induce DNA damage through a pro-oxidant mechanism, indicating a genotoxic risk even at very low micromolar concentrations [10]. 

Several strategies have been used in the past years to improve the stability of *trans*-resveratrol by using different formulations and stabilizing agents [11,12]. To the best of our knowledge, to date there are no works in the literature that evaluate the photo-protection of *trans*-resveratrol in terms of degradation and concomitant THP formation after UV-light exposure [13,14,15,16,17,18,19]. The aim of this work was to improve *trans*-resveratrol photostability by applying an innovative *green* chemistry approach via the use of natural deep eutectic solvents (NaDES) and to inhibit the conversion into the *cis*- isomer and subsequent THP formation.

NaDES comprise a mixture of a hydrogen bond acceptor (e.g., choline, betaine, proline) and a hydrogen bond donor (polyols or natural plant-based organic ions, such as amino acids, carboxylic acids, sugars) in a solid (or solid/liquid) state that associates by hydrogen bonding [20,21,22]. The resulting mixture is eutectic—i.e., it has a lower melting point than that of each individual component and is liquid even at very low temperatures. In addition to being completely biocompatible and eco-sustainable, these eutectic solvents demonstrate an excellent capacity for solubilizing organic compounds. Different NaDES comprising choline, proline or betaine as hydrogen acceptors, and glycerol or propylene glycol as hydrogen donors, were prepared and tested for their photo-stabilizing effect on *trans*-resveratrol subjected to UV-light irradiation (Figure 2). 

The determination of resveratrol and its photoconversion products was improved by the use of a novel chromatographic method, employing for the first time a non-commercial reverse phase column. Several chromatographic methods for the quantification of *trans*-resveratrol in different biological and pharmaceutical matrices are present in the literature [23,24,25,26,27,28]. Conversely, the determination and quantification of *cis*-resveratrol is more complicated due to its low stability and the lack of an analytical standard. However, it is possible to quantify this compound by the induced photo-generation of the *cis*-standard in situ, starting by irradiating the *trans*-isomer and analyzing the sample in the timeframe in which only the isomerization process occurs [23,29,30,31]. As regards the photo-chemical toxic derivative THP, whose analytical standard is not commercially available, no liquid chromatographic methods for the separation and quantification of this compound are present in the literature. In our previous works, we described the HPLC analytical determination and semi-preparative purification of THP on a C18 reverse phase column [9,12]. In the present work a novel near UHPLC method was set up for the optimized chromatographic separation and determination of *trans*-resveratrol, *cis*-resveratrol and THP by using a non-commercial sub-3 µm pentafluorophenyl (PFP) silica-based stationary phase. This method was applied to assess the capacity of NaDES to act as *trans*-resveratrol protecting agents against photo-chemical induced degradation, with a particular focus on the inhibition of THP formation.

## 2. Results and Discussion

### 2.1. Determination of the Photochemical Reaction Products

A representative chromatographic separation of the three compounds of interest (*trans*-resveratrol, *cis*-resveratrol and THP) using the PFP stationary phase is shown in Figure 3. *trans*-Resveratrol standard solution was analyzed before and after one hour of direct sunlight exposure.

The black chromatogram represents the chromatographic profile of a *trans*-resveratrol standard solution, whereas the red chromatogram is the chromatographic profile of a *trans*-resveratrol solution exposed to the sunlight for 1 h. In this latter chromatogram, the first two eluting peaks correspond to *trans*- and *cis*-resveratrol, identified by means of their retention times and spectrometric features. The UV-visible and mass spectrum of both isomers are in agreement with the structure of the molecules and with previously published experimental data. Resveratrol isomers (as isobars) present the same pseudo-molecular ion in their mass spectrum at 227 *m*/*z* [9,31] (Figure 4).

The third eluting peak is represented by the photo-induced electrocyclization product THP, previously synthesized and characterized by our group [9]. The spectral features of this product are shown in Figure 5.

THP is formed via photo-induced *cis*-resveratrol ring closure and subsequent oxidation to give rise to the completely aromatic phenanthrenic system. The oxidative dehydrogenation involves the formal loss of 2 Da, as confirmed by the mass spectrum of THP with respect to that of resveratrol isomers (pseudomolecular ions at 227 *m*/*z* for *trans*/*cis*-resveratrol and 225 *m*/*z* for THP).

This optimized chromatographic method offers a complete resolution of the three photochemical products derived from the *trans*-isomer UV-light irradiation. The use of the pentafluorophenyl stationary phase offers better results with respect to common C18 reverse-phase columns reported in the literature [25]. One of the most important advantages compared with the previously reported methods is the complete separation of *cis*-resveratrol from the phenanthrenoid species THP (α = 1.32). The better selectivity of pentafluorophenyl stationary phase towards these two compounds could arise from the stronger π–π interaction that the tricyclic THP system could exert with the aromatic moiety of the perfluorinated phenilic stationary phase. 

In our previous work, we described the HPLC analytical determination and semi-preparative purification of THP on a C18 reverse phase column [9]. In this work we have optimized the chromatographic separation by using a different UHPLC stationary phase, allowing the determination of the three chemical species in a single chromatographic analysis. The use of a sub-3 μm column with this specific geometry, in addition to the improvement in the chromatographic efficiency, significantly reduces the analysis time by working under isocratic conditions (5 min separation in 30% aqueous CH_3_CN) and avoiding column washing and re-equilibration.

### 2.2. Effect of NaDES on trans-Resveratrol Photochemical Conversion

Figure 6 shows the effect of different NaDES on *trans*-resveratrol UV-light-induced photochemical conversion.

The *trans*/*cis* photo-induced isomerization process is strongly inhibited when *trans*-resveratrol is irradiated in the presence of glycerol containing NaDES. In particular, ProGlc and BetGlc were revealed to be the best solvents in terms of isomerization inhibition. The control solvent (DMSO) displays a similar behavior to that of propylene glycol containing NaDES with a rapid conversion of *trans*- into *cis*- isomer within the first 30 min of analysis. ProGlc and BetGlc showed a pronounced effect also on THP production over two hours of UV-light exposure. The photo-induced production of THP turned out to be less than 5% in the presence of ProGlc NaDES, while it gradually increased in the other NaDES and more than 20% in the control solvent after two hours of UV-light exposure. 

Given the results obtained, three representative samples were exposed to direct sunlight in order to verify the photo-conversion trend in a system that was much closer to environmental conditions. Figure 7 shows the photo-degradation rate of *trans*-resveratrol dissolved in two glycerol-containing NaDES (ProGlc and CholGlc) and in the control solvent (DMSO).

The results are in good agreement with the previous data obtained by the UV-lamp irradiation. *trans*-Resveratrol exposure to direct sunlight radiation results in a rapid almost complete isomerization into the *cis*-form within one hour of irradiation and up to 30% THP conversion after two hours. In the presence of both NaDES, these values are significantly reduced, particularly when *trans*-resveratrol is dissolved in ProGlc eutectic solvent. Notably, after two hours of exposure in ProGlc, the *cis/trans* isomerization was remarkably lower than in the control sample and THP formation was around 2%, i.e., more than ten times lower than control in which was around 30%. The CholGlc solvent also showed an important photoprotective effect, with only a 60% *trans*-resveratrol conversion after two hours of exposure and less than 15% of THP production.

Photochemical isomerization and electrocyclization/oxidation are not the only processes that can occur during resveratrol irradiation. To verify the protective effect on *trans*-resveratrol and to correlate it with the photoisomerization and cyclization trend, the absolute amount of resveratrol (mg/mL) in the two different conditions after 2 h irradiation was calculated. In Table 1 the absolute amount of *trans*-resveratrol in DMSO and ProGlc is reported. As shown in the table, the absolute amount of *trans*-resveratrol present under the two conditions is significantly higher in the presence of ProGlc NaDES.

In all experimental conditions, the total amount of the three compounds (*trans*/*cis*-resveratrol and THP) was assessed to be ≥90% of the starting amount of irradiated material. Our data confirmed the photo-protective effect of ProGlc NaDES on *trans*-isomer photo-chemical degradation and evidenced that isomerization and electrocyclization processes represent one the main photochemical degradation pathways under irradiation.

It should be emphasized that the protective effect of NaDES on the photochemical degradation of *trans*-resveratrol is more pronounced in the presence of glycerol-based solvents with respect to propylenglycol-based ones or of the control solvent under all irradiation conditions and times of treatment. Figure 8 represents the trend of the sequential photochemical reactions of resveratrol in propylene glycol or glycerol based NaDES (i.e., ProProp and ProGlc) compared with DMSO as a control solvent. 

The kinetics of *trans*-resveratrol photoconversion in DMSO were in agreement with the data reported in the literature. The photochemical UV-light irradiation of resveratrol in organic/hydroalcholic solutions causes π–π * transitions that lead to a photostationary state significantly richer in the *cis*- isomer (as shown in Figure 8 for DMSO and ProProp up to one hour irradiation). Conversely, in glycerol-based NaDES (e.g., ProGlc), there is an inversion of the isomerization equilibrium in the photostationary state, reaching a condition in which the *trans*- isomer is significantly more stable than the *cis*- one. The highest stabilization of the *trans*-isomer can in part also explain the significant lower production of THP in NaDES (Figure 9).

The inversion of isomerization equilibrium could be probably unrelated to the physico-chemical properties of the solvents (e.g., viscosity, dielectric constant) but in part may be due to solvation (internal van der Waals stabilization for *cis*-stilbene is lower when solvated) and more probably to hydrogen bond donor effect of resveratrol in the eutectic system with respect to common solvents. It is interesting to note how glycerol and propylene glycol differ only for one hydroxyl moiety. The photoconversion kinetics of resveratrol in common organic solvents including alcohols such as ethanol or propanol are very similar to those observed in DMSO. The observed photo-stabilizing effect of glycerol-containing NaDES could be unrelated to the polarity of the solvent or to the solvation of the molecule itself, instead being related to the ability of the resveratrol to make hydrogen bonds in the eutectic solvent and in this way to maintain a more rigid conformation that increases the isomerization energy barrier [7,32,33]. It is possible that resveratrol in this case could behave as a hydrogen bond donor, becoming an additional component of the eutectic mixture. Future studies on resveratrol and NaDES interaction will be performed for the investigation of the molecular interaction behind this photo-chemical stabilizing effect.

## 3. Materials and Methods

### 3.1. Chemicals and Reagents

Gradient grade methanol, acetonitrile and ultrapure water for UPLC were purchased from Carlo Erba (Milan, Italy). All other chemicals and solvents were of analytical grade and were from Sigma-Aldrich (Milan, Italy). THP was photochemically synthesized and purified in our laboratory as previously described [9].

### 3.2. NaDES Preparation

NaDES were prepared by mixing two components at 70 °C in a water bath under magnetic stirring for 1 h. Six different NaDES were prepared with specific molar ratios after empirical experimental optimization. Choline/glycerol (1:1.5 molar ratio) (CholGlc); proline/glycerol (1:2 molar ratio) (ProGlc); proline/propylene glycol (1:4 molar ratio) (ProProp); betaine/propylene glycol (1:4 molar ratio) (BetProp); betaine/glycerol (1:2.2 molar ratio) (BetGly); choline/propylene glycol (1:3 molar ratio) (CholProp).

### 3.3. Samples Preparation and UV-Light Irradiation

*trans*-Resveratrol was dissolved at a concentration of 10 mg/mL both in DMSO and in the NaDES. All the solutions were extensively sonicated in an ultrasonicator bath to allow the complete dissolution of the molecule, and were carefully protected from direct light. After samples preparation, 0.5 mL of each solution was spread on a glass slide and exposed to UV-lamp irradiation (20 cm from the irradiation source, with a 14.7 W UV-B fluorescent tube emitting at wavelengths of 270−320 nm with a peak at 312 nm) or to direct sunlight (solar light spectrum) for different times. At each time point, a known amount of each sample was diluted in 30% aqueous acetonitrile at a final concentration of 0.2 mg/mL for subsequent chromatographic analysis.

### 3.4. Ultra-High Performance Liquid Chromatography and Mass Spectrometry

UPLC-DAD/MS was performed on a Waters Acquity H-Class UPLC system (Waters, Milford, MA, USA), including a quaternary solvent manager (QSM), a sample manager with a flow through needle system (FTN), a photodiode array detector (PDA) and a single-quadruple mass detector with electrospray ionization source (ACQUITY QDa). Chromatographic analyses were performed on a non-commercial pentafluorophenyl column (75 mm × 3.2 mm i.d., 2.5 μm particle size) developed and prepared by Ciogli and co-workers [34]. The mobile phase was composed by solvent A, 0.1% aqueous HCOOH, and solvent B as 0.1% HCOOH in CH_3_CN. The flow rate was 0.8 mL/min, the column temperature was set at 25 °C and the elution was performed isocratically with 30% B. Samples were diluted in the mobile phase and injected through the needle. The PDA detector was set up in the range 200 to 600 nm. Mass spectrometric detection was performed in the negative electrospray ionization mode, using nitrogen as the nebulizer gas. Analyses were performed in the Total Ion Current (TIC) mode with a mass range of 100–500 *m*/*z*. The capillary voltage was 0.8 kV, cone voltage 15 V, ion source temperature 120 °C and probe temperature 600 °C. Quantification of each compound was performed by using standard calibration curves in the range of 0.02–5 nmoles.

### 3.5. Resveratrol Isomers and THP Determination 

*trans*-Resveratrol and THP were quantified via UPLC-PDA (254 nm) by using standard calibration curve of the pure compounds. *Cis-* isomer standard was generated photochemically as reported by Francioso and co-workers by exposing hydro-alcoholic solutions of *trans*-resveratrol to UV light (20 cm from the irradiation source, with a 14.7 W UV-B fluorescent tube emitting at wavelengths of 270−320 nm with a peak at 312 nm). The identity of the compounds was confirmed by chromatographic retention properties, UV-visible and MS spectra. The conversion rate was estimated by differential calculation between quantified *trans*- isomer after the treatment with UV light (exposure time, 2 min). The calibration curves were obtained by plotting the peak-area ratio of each analyte versus its concentration (*trans*-, R^2^ = 0.9999; *cis*-, R^2^ = 0.9995; THP, R^2^ = 0.9999). The relative percentage of each compound in the reaction mixture was calculated by expression as 100% of the sum of three compounds present at each time of UV-light exposure. Repeatability of the analysis for the three compounds was assessed as lower than 5% expressed (RSD) for standard solutions and irradiated samples.

## 4. Conclusions

This paper describes an improved liquid chromatographic method for the separation of *trans*-resveratrol from its photochemically induced conversion and transformation products. In particular, the method is highly powerful for the determination of resveratrol isomers and THP phenanthrenic photoproducts. THP is a polycyclic aromatic hydrocarbon derived from *trans*-resveratrol photo-chemical isomerization and subsequent electrocyclization followed by oxidation. This product was isolated and identified for the first time in our laboratory, and its chemical structure was elucidated by means of spectroscopic and spectrometric techniques [9]. This photo-derivative is associated with a strong genotoxic risk due to its polycyclic aromatic nature. In another recent work we characterized the type of DNA damage that was demonstrated to be due to a pro-oxidant mechanism [10].

Nowadays, resveratrol is widely used in cosmetic and pharmaceutical topical formulations that are often in contact with the skin and exposed to sunlight UV radiation. Despite its wide use and the many studies aimed at improving resveratrol photostability, to date there are no works in which the production of THP has been evaluated under conditions of different UV light exposure. We therefore applied our improved chromatographic method to evaluate the effect of NaDES on the photoconversion and THP production after *trans*-resveratrol exposure to UV light.

Our results demonstrate a pronounced photoprotective effect exerted by some NaDES under different irradiation conditions (UV-lamp light and direct sunlight exposure). In particular, it was shown that NaDES containing glycerol as a hydrogen bond donor have a strong inhibitory effect on the photo-induced *trans*/*cis* resveratrol isomerization process and a strong reducing effect on the production of the toxic derivative THP.

These data provide new insights for the development of green eco-friendly and biocompatible resveratrol photo-stable pharmaceutical formulations. Furthermore, this work will pave the way for future more in-depth studies on the interactions and molecular mechanisms behind the photochemical stabilization of resveratrol by natural deep eutectic solvents.

## Figures and Tables

**Figure 1 molecules-27-02348-f001:**
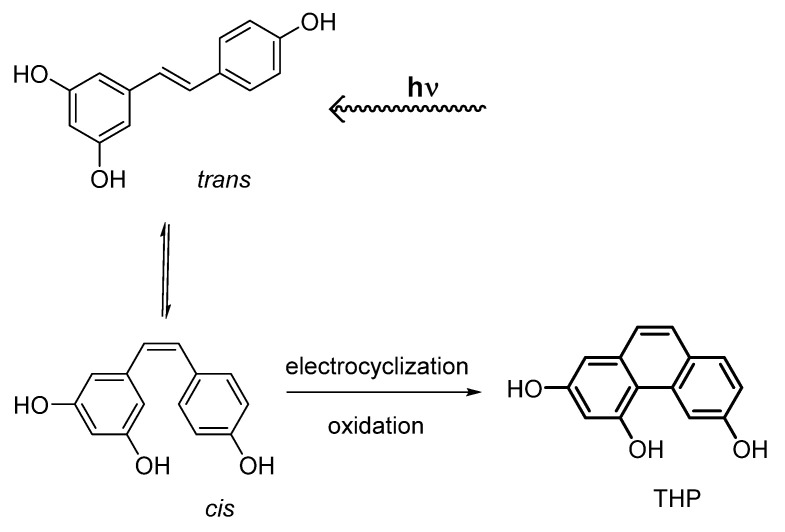
*trans*-Resveratrol photochemical isomerization and electrocyclization/oxidation.

**Figure 2 molecules-27-02348-f002:**
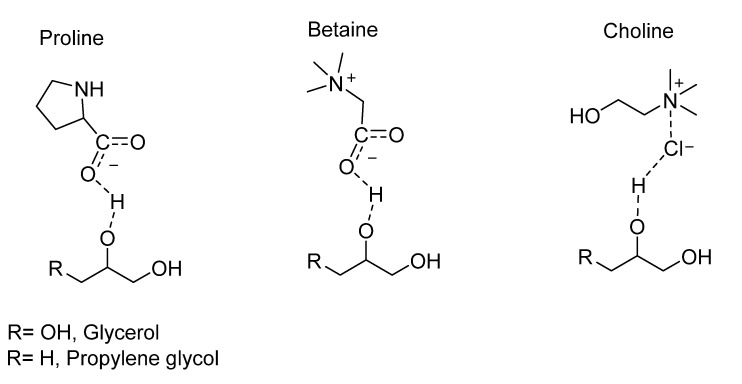
NaDES prepared and evaluated in this work. From the left: proline-, betaine- and choline-based NaDES as hydrogen bond acceptor and glycerol (R=OH) or propylene glycol (R=H) as hydrogen bond donors.

**Figure 3 molecules-27-02348-f003:**
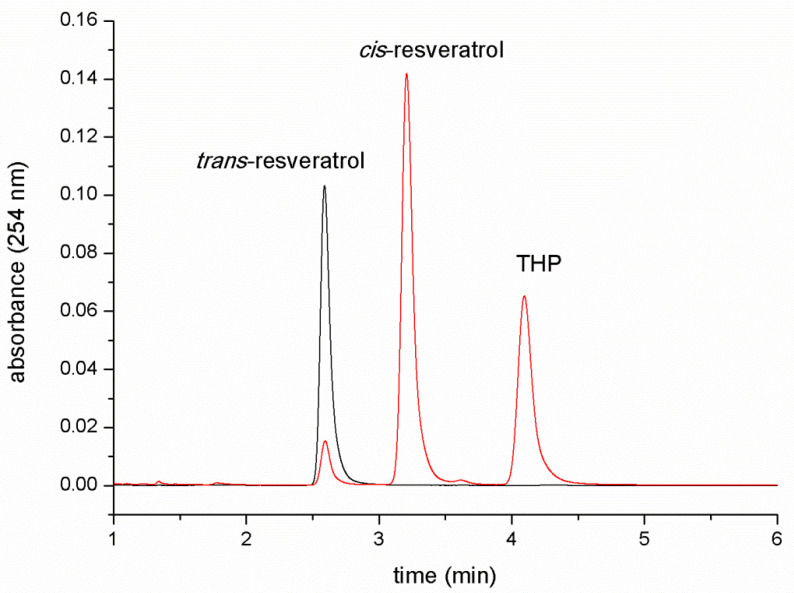
Chromatographic separation of resveratrol isomers and THP before (black chromatogram) and after (red chromatogram) sunlight exposure of a 10 mg/mL solution of the *trans*-isomer in DMSO for 1 h. Retention times were 2.6 min for *trans*-resveratrol, 3.2 min for *cis*-resveratrol and 4.1 min for THP.

**Figure 4 molecules-27-02348-f004:**
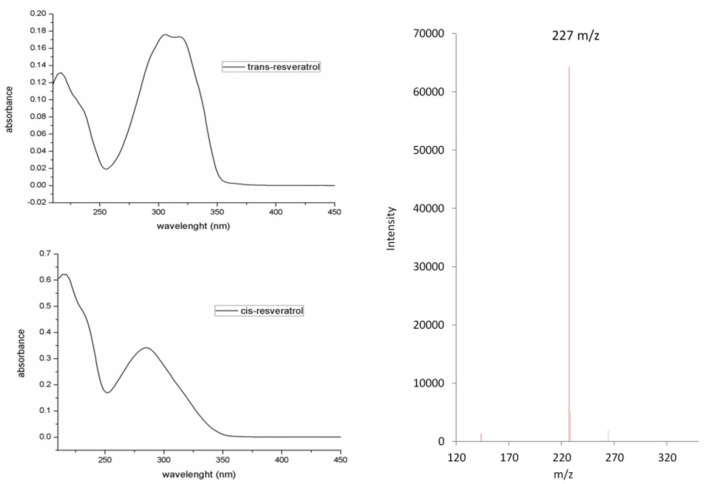
*t**rans*- and *cis*-Resveratrol UV-visible spectra (**left**) and ESI-MS negative mode spectrum (**right**, same pseudo-molecular *m*/*z* value for *cis*- and *trans*- isobars).

**Figure 5 molecules-27-02348-f005:**
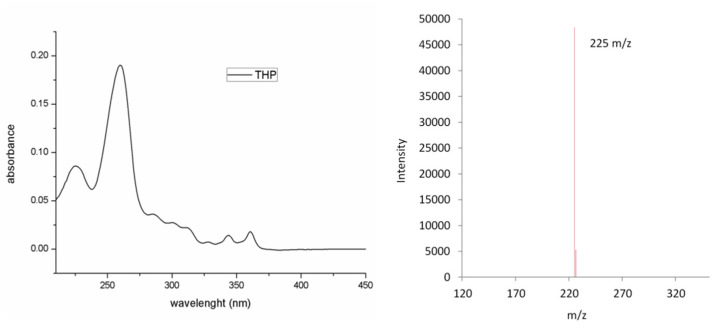
UV-visible (**left**) and ESI-MS negative mode spectrum of THP (**right**).

**Figure 6 molecules-27-02348-f006:**
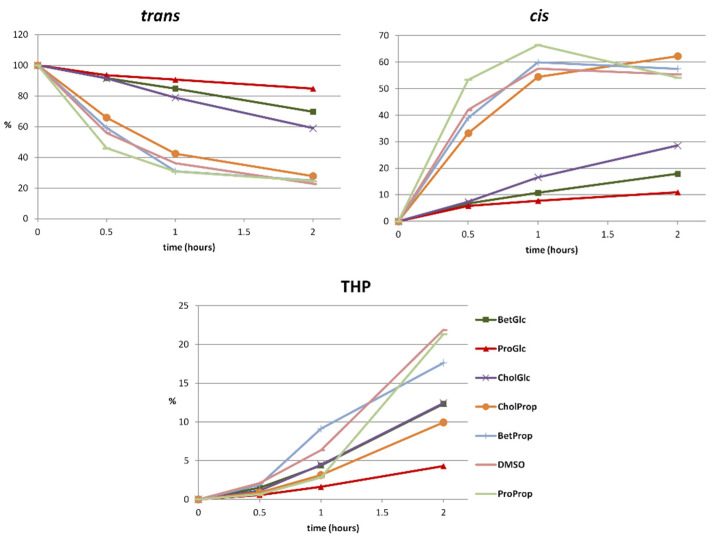
Resveratrol UV-light-induced photochemical isomerization and THP formation in the presence of different NaDES. DMSO was utilized as the control solvent.

**Figure 7 molecules-27-02348-f007:**
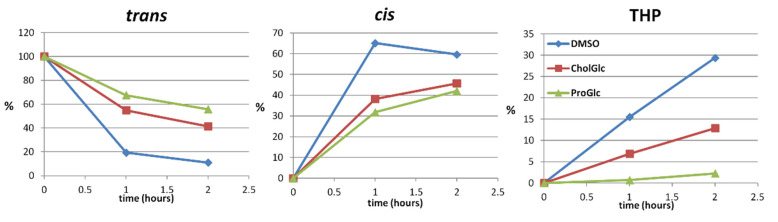
Resveratrol sunlight-induced photochemical isomerization and THP formation in the presence of different NaDES. DMSO was utilized as the control solvent.

**Figure 8 molecules-27-02348-f008:**
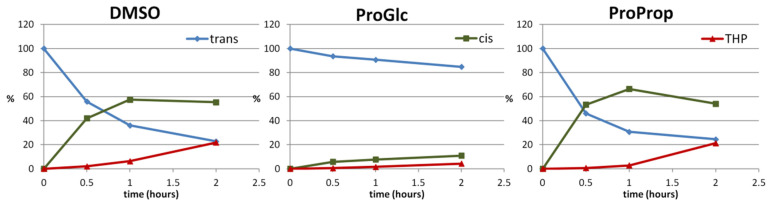
Kinetics of the photochemical reactions of *trans*-resveratrol UV-light irradiation in different solvents.

**Figure 9 molecules-27-02348-f009:**
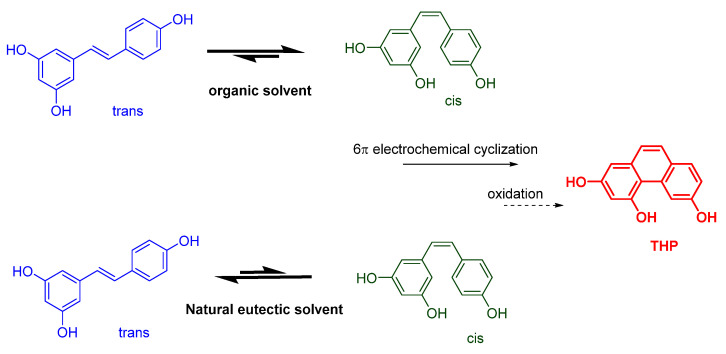
Photochemical equilibrium during *trans*-resveratrol UV-light irradiation. The *trans*/*cis* equilibrium in the photostationary state is generally shifted to the right for resveratrol in organic solvent solution (upper part of the graph). On the contrary, in glycerol-based NaDES solutions, the equilibrium is shifted to the left stabilizing the *trans*- isomeric form (lower part of the graph).

**Table 1 molecules-27-02348-t001:** *trans*-Resveratrol degradation after UV-light irradiation.

*trans*-Resveratrol	Sunlight Exposure (mg/mL)		UV-Light Lamp Exposure (mg/mL)	
Irradiation time (minutes)	DMSO	ProGlc	DMSO	ProGlc
0	9.9	9.8	10.1	9.9
120	1.2	4.3	1.9	7.5

RSD ≤ 5%.

## Data Availability

Not applicable.

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
