# Peer review of "Effect of Natural Deep Eutectic Solvents on trans-Resveratrol Photo-Chemical Induced Isomerization and 2,4,6-Trihydroxyphenanthrene Electro-Cyclic Formation"

_molecules, 2022, doi:10.3390/molecules27072348_

Round 1
Reviewer 1 Report
Mattioli et al. have investigated the effect of Natural Deep Eutectic Solvents (NaDES) as possible photo-protective agents of trans-resveratrol photodegradation when it is exposed to UV-radiation. For this purpose, the authors evaluated the cis-resveratrol isomer and 2,4,6-trihydroxyphenanthrene (THP) formation using HPLC and mass spectrometry. Furthemore, the authors have improved an efficient chromatographic separation method of cis-resveratrol, trans-resveratrol and THP by using a non-commercial pentafluorophenyl stationary phase. The results obtained in this work are important from a biomedical point of view, as well as, for the farmaceutic and cosmetics industry. The experiments have been well designed and the experimental data have been well interpreted. Nevertheless, I do not recommend the manuscript for publication in Molecules. In my opinion, this work is more suitable for the Chemistry or Photochem Journals. Besides, in order to improve the manuscript, I recommend that the following points be taken into account:
1) Taking into account that the photodegradation of resveratrol is studied in this work and, in order to compare whether the irradiation conditions used in the laboratory are similar to those which take place under sunlight conditions, the visible irradiance and employed doses must be specified for both irradiation conditions. The wavelength of excitation of UV lamp should be specified. The authors mentioned that the solution is spread on a glass slide and exposed to irradiation. After that the irradiated sample is diluted to be analyzed by HPLC. What can the authors say about the reproducibility of this form of irradiation and manipulation of the sample? Error bars should be added to each experimental result.
2) The photodegradation of trans-resveratrol is strongly inhibited when it is irradiated in the presence of glycerol(C3H8O3) containing NaDES, being Proline:Glycerol (ProGlc) and Betaine:Glycerol (BetGlc) the best solvents in term of photoinhibition. When propylene glycol (CH3CH(OH)CH2OH) containing NaDES is used as a solvent a negligibly photoprotective effect was observed. Taking into account that the difference between glycerol y propylene glicol is an OH group, the authors should discuss in more depth the different results obtained when glycerol or propylene glycol are used. Did the authors perform experiments using propanol?
3)The authors said concerning Figure 5:”The photo-induced production of THP, turned out to be less than 5% in the presence of these two NaDES, while it gradually increases in the other NaDES and up to 20% in the control solvent after two hours of UV-light exposure”. This is wrong; the percentage of THP production for BetGlc is higher than 10%. Furthemore, the sum of cis, trans and THP percentage is higher than 100%. For this reason, to make a better interpretation of the results an error bar should be added to each curve.
Author Response
Dear reviewer,
Thanks for your suggestions and comments. We found your critical revision very helpful. We improved all the work in different parts. In particular the modifications according to your suggestions are marked in yellow.
As regards in particular
point 1)
We added in the method and materials section more details. The wavelength of excitation of UV lamp was added. We specified in the methods that the repeatability of the process was lower than 5% (expressed as RSD) for standards and for irradiated samples. We also added a table in the results section in which the absolute amount of trans-resveratrol was calculated.
Point 2)
We take into consideration your very interesting question and as for you request we improved the discussion section. Glycerol and propylene glycol as you said differ just for one OH group and this is a very interesting thing to deepen discuss (a figure with the structure of NaDES was added in the introduction section for more clarity). Regarding your question about the use of isopropanol, in our previous work we analysed and purified THP from trans-resveratrol irradiated solution (Francioso et al . J. Org. Chem. 2014, 79, 19, 9381–9384) and in that case we used isopropanol as solvent even to favourite the solubilization of THP as final product. We observed that the photoconversion kinetics of resveratrol in common organic solvents and alcohols is very similar but we never observed an inversion of isomerization equilibrium as in this new reported case in presence of NaDES. We discuss this results more in depth in the results and discussion section. Thank you again for your interesting questions
Point 3)
Thank you for notice the error. It was our mistake. We modified as your suggested. The kinetics calculations were done for each compound and expressed as relative percentage respect to the sum of the three compounds at each point of analysis. We also maintain the kinetic graphs and specify the repeatability in the text. As said above also the absolute amount of trans-resveratrol was calculated to verify the degradation of the starting material besides the photo-conversion behaviour (table added).
We want to thanks again for the suggestions and all the revision of the paper that we found very useful for the improvement of the quality and the presentation of our manuscript
Best regards
Dr. Antonio Francioso

Reviewer 2 Report
This manuscript describes the protective effect on the photochemical conversion of trans-resveratrol using NaDES prepared with proline and glycerol. Among six NaDESs studies, three (BetGlc, ProGlc, ChoGlc) showed inhibitory activity and the other three (CholProp, BetProp, ProProp) did not when UV-light was used. These results indicate the importance of the glycerol component. Interesting findings are presented, but there is insufficient information on experimental methods. Additionally the discussion on the mechanism is poorly written. Major revisions need to be made for publication in Molecules.
The detailed comments:
- Page 3, line 36: Since "a" does not exist, "a" in "Fig. 2a" should be deleted.
- Page 4, line 1: in the figure ® in the Fig. 2
- Fig 3: It is unclear whether the MS spectrum of trans-resveratrol or cis-resveratrol is given. Both spectra must be shown.
- Quantitative HPLC analysis should be done. Since only the ratio of products is given, it is unclear whether NaDES inhibited isomerization or degradation of resveratrol.
- The wavelength of the UV light must be shown.
- UV spectra of NaDES and resveratrol in NaDES must be measured. Possible suppression mechanisms include wavelength change or absorption of light by NaDES.
- Rational explanation for the relationship between the inversion of isomerization equilibrium and solvation should be provided with citations.
Author Response
Dear reviewer,
Thanks for your suggestions and comments. We found your critical revision very helpful. We improved all the work in different parts. In particular the modifications according to your suggestions are marked in yellow.
point 1) A figure was added for better explanation and presentation of NaDES
Point 2) Done. Figure numbers changed.
Point 3) We clarified and explain the figure as you suggested. The UV spectrum is different for trans- and cis- isomers but the mass spectrum of both molecules displays the same MS pseudo-molecular ion at 227 m/z (as isomers and isobars)
Point 4) We improved methods and material section explaining in more detail. The kinetics calculations were done for each compound and expressed as relative percentage respect to the sum of the three compounds at each point of analysis. As you suggested quantitative calculations for absolute amount of trans-resveratrol were done to verify the degradation of the starting material besides the photo-conversion behaviour (table added).
Point 5) Done according to your suggestion
Point 6) Thank you for this interesting question. The studied NaDES don’t have characteristic UV visible spectrum and do not absorb UV-light. We measured the UV spectrum of all NaDES and as reported, we can confirm that no one of the studied solvent displays UV-light absorption features.
Point 7) The discussion section was improved and rational explanation of all the results was discussed deepen in several parts of the work. Thank again for this comment.
We want to thanks again for the suggestions and all the revision of the paper that we found very useful for the improvement of the quality and the presentation of our manuscript
Best regards
Dr. Antonio Francioso
